# Scene-adaptive Knowledge Distillation for Sequential Recommendation via Differentiable Architecture Search

**Lei Chen**[1], **Fajie Yuan**[2], **Jiaxi Yang**[3], **Chengming Li**[4], **Min Yang**[3*]

[1]The University of Hong Kong     [2]Westlake University
[3]Shenzhen Institute of Advanced Technology, Chinese Academy of Sciences
[4]Shenzhen MSU-BIT University
lchen@cs.hku.hk, yuanfajie@westlake.edu.cn
{jx.yang, min.yang}@siat.ac.cn, licm@smbu.edu.cn

## Abstract

Sequential recommender systems (SRS) have become a research hotspot due to their power in modeling user dynamic interests and sequential behavioral patterns. To maximize model expressive ability, a default choice is to apply a larger and deeper network architecture, which, however, often brings high network latency when generating online recommendations. Naturally, we argue that compressing the heavy recommendation models into middle- or light-weight neural networks that reduce inference latency while maintaining recommendation performance is of great importance for practical production systems. To realize such a goal, we propose AdaRec, a knowledge distillation (KD) framework which compresses knowledge of a teacher model into a student model adaptively according to its recommendation scene by using differentiable neural architecture search (NAS). Specifically, we introduce a target-oriented knowledge distillation loss to guide the network structure search process for finding the student network architecture, and a cost-sensitive loss as constraints for model size, which achieves a superior trade-off between recommendation effectiveness and efficiency. In addition, we leverage earth mover's distance (EMD) to realize many-to-many layer mapping during knowledge distillation, which enables each intermediate student layer to learn from other intermediate teacher layers adaptively. Extensive experiments on three real-world recommendation datasets demonstrate that our model achieves significantly better accuracy with notable inference speedup compared to strong counterparts, while discovering diverse architectures for sequential recommendation models under different recommendation scenes.

## 1 Introduction

Sequential (a.k.a. session-based) recommender systems (SRS) that aim to predict new interactions based on user historical ones have attracted much attention in recent years [7, 34, 44, 12, 31, 41, 47, 19]. In particular, with the tremendous success of deep learning, deep neural network (DNN) based sequential recommendation (SR) models have yielded substantial improvements comparing to traditional collaborative filtering (CF) [7], such as neighborhood methods [30] and shallow factorization models [13]. This is because with many hidden layers, well-designed deep models could be more powerful in capturing user dynamic interests, high-level or long-range sequential relations of

---

*Min Yang is the corresponding author.

Workshop on Advancing Neural Network Training at 37th Conference on Neural Information Processing Systems (WANT@NeurIPS 2023).

user interactions. More recently, Chen *et al.* [3] revealed that highly expressive deep SR models such as NextItNet [44], SASRec [12] and BERT4Rec [31] could be stacked in a surprised depth with over **100 layers** for achieving their optimal performance.

However, a real problem arises as these deep SR models go bigger and deeper; that is, the model becomes too large in parameter size, and both memory and inference costs increase sharply, making the deployment of them difficult in production systems. Thereby, we argue that compressing the heavy deep SR models into moderate- or light-weight neural networks without sacrificing their accuracy is of crucial importance for practical usage. Knowledge distillation (KD) [8] as an effective compression technique has been recently investigated in the recommender systems domain [35, 25, 10]. By transferring useful knowledge from a big teacher network to the student network, large deep models could be slimmed into a smaller and shallower structure without performance degradation. However, existing KD methods basically distill the teacher model into a fixed-structure student model that is manually designed in advance. This potentially limits the flexibility and scalability of the student model, especially for diverse and relatively complicated scenarios in recommender systems. For example, the optimal structure for music recommendation might be different from the optimal structure for E-commerce recommendation. Ideally, we hope to build an adaptive student model whose optimal structure takes full consideration of the specific recommendation scenarios.

Inspired by the success of automated machine learning (AutoML), we propose a novel KD method to compress the deep SR models, termed as AdaRec. AdaRec distills the knowledge of a teacher model into a student model adaptively according to the recommendation scene based on differentiable neural architecture search (NAS) [21, 40, 2, 48]. Specifically, we devise a target-oriented KD loss to provide search supervision for learning the architecture of student network, and a cost-sensitive loss as additional regularizer to constrain the model size, which achieve a superior trade-off between recommendation effectiveness and efficiency. In addition, we leverage earth mover's distance (EMD) to realize effective many-to-many layer mapping during the distillation process, enabling each intermediate layer of student to learn from any other intermediate layers of its teacher. It is worth noting that, our method is a generic KD framework which can directly apply to a broad class of well-known deep SR models, such as NextItNet [44] and SASRec [12]. In addition, with the well-designed NAS architecture, our method can distill the deep SR models into effective smaller models with diverse network architectures, according to the specific recommendation scenarios.

Our main contributions in this paper are fourfold:

- To the best of our knowledge, we are the first to consider combining KD and NAS in the SRS tasks so as to compress many advanced deep SR models adaptively according to their recommendation scenes.

- We devise a KD loss based on EMD and a cost-sensitive constraint to achieve a trade-off between recommendation effectiveness and efficiency.

- AdaRec is model-agnostic and potentially applicable for any SR model with a deep network architecture. We verify the universality of the AdaRec framework by performing KD with two well-known teacher models, namely, NextItNet [44] and SASRec [12].

- We conduct extensive experiments on three real-world recommendation datasets with different scenarios (E-commerce, music and movie recommendation), demonstrating that AdaRec achieves significantly better accuracy with notable inference speedup comparing to its original teacher model. Moreover, we discover diverse neural architectures of the student model in different recommendation scenarios or tasks.

## 2  Related Work

### 2.1  Deep Sequential Recommendation

SRS is an important branch in the recommendation field and has become a hotspot recently due to the wide range of application scenarios and huge commercial values. Noticeably, DNN have achieved superior recommendation accuracy in SRS tasks due to their powerful capacity in modeling complicated and long-term user behavior relations. In general, these models could be classified into three categories, namely recurrent neural network (RNN) based, convolutional neural network (CNN) based and self-attention (SA) based methods. RNN has shown superb performance in many natural

language processing (NLP) tasks and were also successfully applied into the SRS field. Specifically, Hidasi *et al.* [7] proposed GRU4Rec, which is the first RNN-based SR model. While effective, RNN-based SR models rely heavily on the hidden states of the entire past, which cannot take full advantage of the parallel processing resources (e.g., GPU and TPU) [44] during training. Therefore, CNN-based and self-attention based models are proposed to mitigate such limitations [34, 44, 12, 31, 50]. Among them, Tang *et al.* [34] proposed Caser, which embeds a sequence of user-item interactions into an "image" and learns sequential patterns as local features of the image by using wide convolutional filters. Subsequently, [44] proposed NextItNet, a very deep 1D temporal CNN-based recommendation model which particularly excels at modeling long-range item sequences [33, 43, 45]. In addition, self-attention based models, such as SASRec [12] and BERT4Rec [31], also showed competitive accuracy for SRS tasks. SASRec [12] utilized the popular self-attention mechanism to model long-term sequential semantics by encoding user's historical behaviors. Inspired by the great success of BERT [4] in NLP filed, Sun *et al.* [31] proposed BERT4Rec, which uses the transformer architecture and masked language model (MLM) to learn bidirectional item dependencies for better sequential recommendations. In this paper, we present AdaRec by applying NextItNet and SASRec as teacher networks given their superior performance and very deep or wide network architectures [33, 3] in literature. With the advancement on graph neural networks (GNN), GNN-based SR models, such as SR-GNN [39], GC-SAN [42] and SGL [38], have also attracted attention and yielded substantial improvements in recommendation accuracy. However, given their shallow neural architectures, we simply ignore AdaRec on them.

## 2.2 Knowledge Distillation

Large and deep neural networks have achieved remarkable success in recent recommendation literature [33, 3, 41]. However, the deployment of such heavy model for real production system remains a great challenge. KD [8, 17] is a representative technique for model compression and acceleration. Its basic idea is to transfer important knowledge from a big teacher network to a small student network. Specifically, Tang *et al.* [35] proposed the first KD technique for learning to rank problems in recommender systems. However, the work only focused on distillation on very shallow neural recommendation models while its effectiveness for deep SRS keeps largely unknown. [20] presented a general KD framework for counterfactual recommendation with four types of distillation, namely, label-based, feature-based, sample-based and model structure-based distillation. More recently, [10] proposed a KD framework that forces the student network to learn from both the teacher's output and the latent knowledge stored in the teacher model. In addition, KD-based compression have also been widely studied in other domains [32, 29, 9, 17, 16, 11]. Recently, compressing pre-trained language models (e.g., BERT) with KD has attracted attentions as well, and many novel models are proposed to effectively distill BERT from different perspectives (e.g., embedding layer, hidden layers and prediction layer), such as PKD-BERT [32], DistilBERT [29], TinyBERT [9] and BERT-EMD [17].

## 2.3 Neural Architecture Search

NAS that automatically discovers the network architecture, has gained increasing attention recently. Early NAS methods based on reinforcement learning (RL) [51] and evolution [27] are computationally very expensive. Recent studies significantly speed up the search and evaluation stages by architecture parameter sharing, such as ENAS [26], gradient-descent based DARTS [21, 2] and SNAS [40], and hardware-aware optimization such as AMC [6] and FBNet [37, 36]. Different from existing work, we devise a target-oriented KD loss to provide search supervision for learning the architecture of the student network, which is a joint search of student structure and knowledge transfer under the guidance of the teacher model. To our best knowledge, we are the first to propose a combination of KD and NAS for compressing the deep SR models.

## 3 Our Method

We introduce a novel scene-adaptive KD-based model compression approach with differentiable NAS, called AdaRec. Formally, suppose that a large teacher model $\mathcal{T}$ is trained on a target dataset $D$, and the architecture searching space is denoted as $\mathcal{A}$. The goal of AdaRec is to automatically find a high-performing student model $\mathcal{S}$ from $\mathcal{A}$ with a small number of learning parameters.

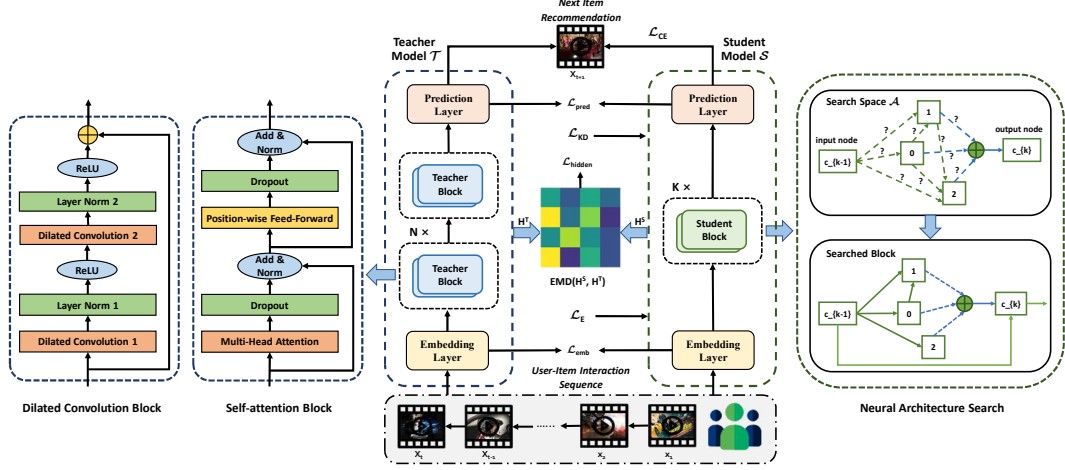

Figure 1: Model architecture of AdaRec. The proposed AdaRec consists of two primary components: a teacher model and a student model.

Figure 1 illustrates the overview of the AdaRec framework. The basic idea is to distill knowledge from a heavy teacher recommendation model $\mathcal{T}$ to a small student model $\mathcal{S}$ adaptively subject to the specific recommendation task. By using such a supervisions from the teacher, the student can achieve comparable performance to the teacher with faster inference time. In this paper, we specify AdaRec using NextItNet [44] and SASRec [12] as the teacher models given their superior recommendation performance. It is noteworthy that the "teacher" is model-agnostic and potentially applicable for any SR model with a deep network architecture. Specifically, the network structures of the student model are automatically searched based on the NAS techniques. To this end, we devise a KD loss to provide search supervision for learning the architecture of the student model and a cost-sensitive loss function as search regularization to control the model size. In this manner, our AdaRec could achieve an impressive trade-off between recommendation accuracy and computational efficiency for SRS tasks.

In what follows, we describe AdaRec by elaborating the teacher model, the student model, the KD process and the NAS searching process.

## 3.1 Teacher Model

We employ the block-wise (e.g., ResNet [5]) deep networks as the teacher models given their powerful performance in literature. The general framework of the teacher model consists of the bottom embedding layer, hidden layers and the softmax layer. Specifically, each item $x_i$ in the user interaction sequence is converted into an embedding $\mathbf{e}_i$, and correspondingly the interaction sequence could be represented by an embedding matrix $\mathbf{E} = [\mathbf{e}_1 \ldots \mathbf{e}_n]$. Afterwards, we pass $\mathbf{E}$ into the hidden layers, and obtain the final hidden representation $\mathbf{V} \in \mathbb{R}^{t \times d}$ where $d$ denotes the embedding dimension. Finally, we apply a softmax function to predict the output probability of the interested item $x_{n+1}$ as follows:

$$p(x_{n+1}|x_{1:n}) = \mathrm{softmax}(\mathbf{W}_2\mathbf{V} + \mathbf{b}_2) \tag{1}$$

where $\mathbf{W}_2$ and $\mathbf{b}_2$ denote the mapping matrix and the bias term respectively.

In terms of the hidden layers, we use the residual blocks from NextItNet [44] and SASRec [12] for case study, where NextItNet is based on the dilated CNN blocks, while SASRec is based on the self-attention blocks. The residual block structures are depicted in Figure 1.

Regarding the training of the two teacher models, we follow their original paper by optimizing NextItNet and SASRec using the left-to-right autoregressive (AR) method [44, 12].

## 3.2 Student Model

Typical model compression methods usually apply KD to transfer knowledge from the heavy teacher network to the manually designed student network, which rely heavily on the prior knowledge of human experts to design the structure of the student model. In this paper, we automatically search the architecture of the student model using NAS techniques rather than designing a fixed network architecture in advance. In addition to learning from the training data with cross-entropy loss, we devise a scene-adaptive KD loss (see Section 3.3) to learn an effective student model by learning from the teacher model. We also employ an efficiency constraint (see Section 3.4) to explicitly takes the efficiency of the student model into the main objective.

Here, we introduce a block-based micro architecture searching method [26], which discovers an optimal network architecture from the pre-defined operation sets (i.e., search space).

**Search Space**   The search space design is key to the final performance of the searched student model. In this study, the large search space of NAS is modularized into blocks so as to reduce the search complexity, similar to [15]. We merely need to automatically search several block structures, and the whole network architecture can be constructed by repeatedly stacking the searched blocks. In this way, we do not need to learn each block from scratch by sharing the structures of all blocks, and therefore less time is required to learn the best performing student. Specifically, we represent each searched block denoted by $\alpha_c$ as a directed acyclic graph (DAG). Each node within the block represents a latent state $h$, and the edge from the node $i$ to the node $j$ denotes the operation $o_{i,j}$ transforming $h_i$ to $h_j$. For the $k$-th ($k > 1$) searched block, we define an input node $c_{k-1}$ and an output node $c_k$, where the output node is computed by attentively summarizing the intermediate nodes. Formally, suppose $\mathcal{O}$ to be the candidate operations, and there are $M$ intermediate nodes in the topological order, i.e., $o_{i,j} \in \mathcal{O}$ exists when $j \geq 1$ and $i < j$. Hence, we define the search space $\mathcal{A}$ as follows:

$$\mathcal{A} = \alpha_c = [o_{0,1}, o_{0,2}, o_{1,2}, \ldots, o_{i,j}, \ldots, o_{M,M+1}] \tag{2}$$

**Operation Set**   For both of the two teacher (base) models (NextItNet and SASRec), we adopt the same operation set to search the student network architecture. In this paper, we employ lightweight CNN-based operations as candidates due to their superior accuracy and computational efficiency in the SRS literature, compared to RNN [7] and SA [12] based models. Concretely, the candidate operations $\mathcal{O}$ contain four kinds of operations: "*convolution*", "*pooling*", "*skip connection*" and "*zero*" operations. The "*convolution*" operations include the 1D convolution, standard convolutions (without dilation), casual dilated convolutions [43] with kernel size $\{3, 5\}$. Note that the dilated convolution is used to capture long-term dependency information. The "*pooling*" operations include the average pooling and the max pooling with kernel size 3. The "*skip connection*" is utilized for the residual connections. The "*zero*" operation represents the absence of connection between nodes.

### 3.3 Scene-adaptive Knowledge Distillation

We devise a KD constrain with EMD, which uses the the teacher network to guide the network architecture search of the student model.

#### 3.3.1 Embedding Layer Distillation

The prediction accuracy of the SR models, such as NextItNet, can be largely improved by increasing the embedding dimension [33]. Compressing item embedding matrices without reducing the recommendation performance is vital for online inference speedup and parameter reduction. We define $\mathcal{L}_{\text{emb}}$ as the distillation loss of the embedding layer, where it is minimized by the mean squared error (MSE) between the teacher network and the student network:

$$\mathcal{L}_{\text{emb}} = \text{MSE}\left(\mathbf{E}^T, \mathbf{E}^S \mathbf{W}_e\right) \tag{3}$$

where $\mathbf{E}^T$ and $\mathbf{E}^S$ represent the item embedding matrices of teacher and student models, respectively. $\mathbf{W}_e$ is a learnable projection parameter.

#### 3.3.2 Prediction Layers Distillation

The student model is encouraged to match the prediction ability of the teacher model by learning from the probability logits of the teacher. We define $\mathcal{L}_{\text{pred}}$ using Kullback-Leibler (KL) divergence [14] as

the distillation loss of the prediction layer:

$$\mathcal{L}_{\text{pred}} = \text{KL}\left(\mathbf{z}^T, \mathbf{z}^S\right) \tag{4}$$

where $\mathbf{z}^T$ and $\mathbf{z}^S$ are probability logits after passing through the softmax layer of the teacher & student models, respectively.

### 3.3.3 Hidden Layers Distillation

Generally, the teacher model and the student model have different numbers of hidden layers, therefore it is not effective to employ the general one-to-one layer mapping techniques in KD. Here, we employ the EMD [28] algorithm to encourage each student hidden layer to learn from multiple teacher layers adaptively. EMD measures the distance between the teacher network and the student network as the minimum cumulative cost of knowledge transfer [28].

The key idea is to treat the hidden layers as distributions, and the desired transformation makes the teacher and student distributions close. Formally, let $\mathbf{H}^T = \left\{\left(\mathbf{H}_1^T, w_{T_1}^{\mathbf{H}}\right), \ldots, \left(\mathbf{H}_N^T, w_{T_N}^{\mathbf{H}}\right)\right\}$ be the hidden layers of teacher model and $\mathbf{H}^S = \left\{\left(\mathbf{H}_1^S, w_{S_1}^{\mathbf{H}}\right), \ldots, \left(\mathbf{H}_K^S, w_{S_K}^{\mathbf{H}}\right)\right\}$ be the hidden layers of student model, where $\mathbf{H}_i^T$ and $\mathbf{H}_j^S$ represent the $i$-th and $j$-th hidden layer of the teacher and student models, $w_{T_i}^{\mathbf{H}}$ and $w_{S_j}^{\mathbf{H}}$ are corresponding layer weights, $N$ and $K$ represent the number of hidden layers in the teacher and student models, respectively. We define a "ground" distance matrix $\mathbf{D}^{\mathbf{H}} = \left[d_{ij}^{\mathbf{H}}\right]$, where $d_{ij}^{\mathbf{H}}$ represents the cost of transferring the knowledge of hidden states from $\mathbf{H}_i^T$ to $\mathbf{H}_j^S$. We adopt KL divergence to calculate the distance $d_{ij}^{\mathbf{H}}$:

$$d_{ij}^{\mathbf{H}} = \text{KL}\left(\mathbf{H}_i^T, \mathbf{H}_j^S \mathbf{W}_h\right) \tag{5}$$

where $\mathbf{W}_h$ is a learnable projection parameter.

Then, a mapping flow matrix $\mathbf{F}^{\mathbf{H}} = \left[f_{ij}^{\mathbf{H}}\right]$, with $f_{ij}^{\mathbf{H}}$ the mapping flow between $\mathbf{H}_i^T$ and $\mathbf{H}_j^S$, is learned by minimizing the cumulative cost required to transfer knowledge from $\mathbf{H}^T$ to $\mathbf{H}^S$:

$$\text{WORK}\left(\mathbf{H}^T, \mathbf{H}^S, \mathbf{F}^{\mathbf{H}}\right) = \sum_{i=1}^{N} \sum_{j=1}^{K} f_{ij}^{\mathbf{H}} d_{ij}^{\mathbf{H}} \tag{6}$$

subject to the following constraints:

$$f_{ij}^{\mathbf{H}} \geq 0 \quad 1 \leq i \leq N, 1 \leq j \leq K \tag{7}$$

$$\sum_{j=1}^{K} f_{ij}^{\mathbf{H}} \leq w_{T_i}^{\mathbf{H}} \quad 1 \leq i \leq N \tag{8}$$

$$\sum_{i=1}^{N} f_{ij}^{\mathbf{H}} \leq w_{S_j}^{\mathbf{H}} \quad 1 \leq j \leq K \tag{9}$$

$$\sum_{i=1}^{N} \sum_{j=1}^{K} f_{ij}^{\mathbf{H}} = \min\left(\sum_{i}^{N} w_{T_i}^{\mathbf{H}}, \sum_{j}^{K} w_{S_j}^{\mathbf{H}}\right) \tag{10}$$

After we solve the aforementioned optimization problem, an optimal mapping flow $\mathbf{F}^{\mathbf{H}}$ can be learned. Then, we define the EMD by normalizing the work over the total flow:

$$\text{EMD}\left(\mathbf{H}^S, \mathbf{H}^T\right) = \frac{\sum_{i=1}^{N} \sum_{j=1}^{K} f_{ij}^{\mathbf{H}} d_{ij}^{\mathbf{H}}}{\sum_{i=1}^{N} \sum_{j=1}^{K} f_{ij}^{\mathbf{H}}} \tag{11}$$

Finally, the hidden-layer distillation loss (termed as $\mathcal{L}_{\text{hidden}}$) can be defined by the EMD between $\mathbf{H}^T$ and $\mathbf{H}^S$:

$$\mathcal{L}_{\text{hidden}} = \text{EMD}\left(\mathbf{H}^S, \mathbf{H}^T\right) \tag{12}$$

### 3.3.4 Knowledge Distillation Loss

By combining the above three distillation objectives ($\mathcal{L}_{\text{emb}}$, $\mathcal{L}_{\text{pred}}$, $\mathcal{L}_{\text{hidden}}$), we can unify the KD loss $\mathcal{L}_{KD}$ between the teacher and student networks:

$$\mathcal{L}_{\text{KD}} = \mathcal{L}_{\text{emb}} + \mathcal{L}_{\text{pred}} + \mathcal{L}_{\text{hidden}} \tag{13}$$

## 3.4 Efficiency Constraint

We also devise an efficiency constraint, which explicitly takes the efficiency of the student model into the main objective to achieve a trade-off between recommendation effectiveness and efficiency. Specifically, we define a cost-sensitive loss by considering both the parameter size and inference time:

$$\mathcal{L}_E = \sum_{o_{i,j} \in \alpha_c} SIZE\left(o_{i,j}\right) + FLOPs\left(o_{i,j}\right) \tag{14}$$

where $SIZE(\cdot)$ denotes the size of normalized parameters. For each operation, we use $FLOPs(\cdot)$ to denote the number of floating point operations (FLOPs). We summarize the FLOPs of the searched operations to approximate the actual inference time of the student model.

## 3.5 Overall Training Procedure

Following the common paradigms of previous KD methods, we first pre-train the large teacher network. Then, the network architecture of the light student is searched automatically with the guidance of the pre-trained teacher. When searching the student architecture, we combine the KD loss $\mathcal{L}_{\text{KD}}$ and the cost-sensitive loss $\mathcal{L}_{\text{E}}$. In addition, we also need to incorporate the cross-entropy loss ($\mathcal{L}_{CE}$) learned on the training data to help search the student architecture. We define the cross-entropy loss function as follows:

$$\mathcal{L}_{CE} = -\sum_{X^u \in \mathbf{X}} p(x_{t+1}^u) \log p(\hat{x}_{t+1}^u) \tag{15}$$

where $\mathbf{X}$ represents the whole user-item interaction sequences in the training data, $p(x_{t+1}^u)$ is the ground truth distribution for next item prediction and $p(\hat{x}_{t+1}^u)$ is the prediction distribution of the searched student model.

The overall loss function is defined as follows:

$$\mathcal{L} = (1 - \gamma)\mathcal{L}_{CE} + \gamma\mathcal{L}_{KD} + \beta\mathcal{L}_E \tag{16}$$

where $\gamma$ and $\beta$ denote the hyperparameters for balancing the three loss functions.

After finishing the joint searching of the student network architecture and knowledge transfer with the supervision of the pre-trained teacher, we can derive an effective, efficient and adaptive architecture as the compressed SR model by stacking the searched block structures.

### 3.5.1 Differentiable Optimization

It is difficult, if not impossible, to directly optimize the objective function in Eq. (16) by using a brute-force algorithm to enumerate over all candidate operations because of the huge combinatorial searching operations. To resolve such an issue, we model the search operation $o_{i,j}$ as discrete variables (one-hot variables) complying to discrete probability distributions $P_o = \left[\theta_1^o, \ldots, \theta_{|\mathcal{O}|}^o\right]$. Afterwards, we employ the Gumbel-Softmax distribution [24] to convert categorical samples into continuous distributions $y^o \in R^{\mathcal{O}}$ as follows:

$$\mathbf{y}_i^o = \frac{\exp\left[\left(\log\left(\theta_i^o\right) + g_i\right)/\tau\right]}{\sum_{j=1}^{|\mathcal{O}|} \exp\left[\left(\log\left(\theta_j^o\right) + g_j\right)/\tau\right]} \tag{17}$$

where $g_i$ is a random noise drawn from Gumbel(0, 1) distribution, $\tau$ is a temperature coefficient controlling the discreteness of the output vectors $\mathbf{y}^o$. In this way, we can optimize the objectives $\mathcal{L}_{KD}$ and $\mathcal{L}_E$ directly using gradient-based optimizers by using the discrete variable $argmax(\mathbf{y}^o)$ in the forward pass and using the continuous vector $\mathbf{y}^o$ in the back-propagation stage.

# 4 Experimental Setup

## 4.1 Experimental Datasets

We conduct extensive experiments on three real-world SRS datasets from three different domains (scenes): RetailRocket from the E-commerce domain, 30Music from the music domain [23], and MovieLens-2K from the movie domain [1]. The statistics of them are provided in Table 1.

Table 1: Statistics of the three datasets (after pre-processing).

| Dataset | #Users | #Items | #Interactions | #Sequences | Length $t$ |
|---|---|---|---|---|---|
| RetailRocket | 104,593 | 70,012 | 916,421 | 134,241 | 10 |
| 30Music | 27,364 | 138,990 | 2,081,086 | 177,818 | 20 |
| ML-2K | 2,112 | 7,871 | 678,935 | 14,518 | 50 |

Table 2: Overall performance comparison on the three datasets in terms of MRR@$N$ & HR@$N$ ($N$ is set to 5), parameter size (Params) and inference speedup (Speedup). Note that the improvements of AdaRec over all baselines are statistically significant in terms of paired t-test with p-value $< 0.01$.

| Model | RetailRocket | | | | 30Music | | | | ML-2K | | | |
|---|---|---|---|---|---|---|---|---|---|---|---|---|
| | MRR@5 | HR@5 | Params | Speedup | MRR@5 | HR@5 | Params | Speedup | MRR@5 | HR@5 | Params | Speedup |
| GRU4Rec | 0.6952 | 0.7748 | \ | \ | 0.5242 | 0.6438 | \ | \ | 0.4115 | 0.6141 | \ | \ |
| Caser | 0.6489 | 0.7132 | \ | \ | 0.5686 | 0.6312 | \ | \ | 0.4186 | 0.6072 | \ | \ |
| SR-GNN | 0.7038 | 0.7788 | \ | \ | 0.5925 | 0.6909 | \ | \ | 0.4267 | 0.6303 | \ | \ |
| NextItNet | 0.7139 | 0.7817 | 40.28M | 1.00× | 0.6149 | 0.7029 | 74.02M | 1.00× | 0.4453 | 0.6462 | 9.87M | 1.00× |
| KD-NextItNet | 0.7124 | 0.7889 | 8.80M | 1.97× | 0.5969 | 0.6961 | 17.29M | 1.87× | 0.4333 | 0.6388 | 1.16M | 2.20× |
| **AdaRec-NextItNet** | **0.7345** | **0.7964** | **8.66M** | **2.31×** | **0.6343** | **0.7151** | **17.15M** | **2.61×** | **0.4489** | **0.6519** | **1.11M** | **2.78×** |
| SASRec | 0.6982 | 0.7511 | 17.80M | 1.00× | 0.5761 | 0.6437 | 34.70M | 1.00× | 0.4241 | 0.6236 | 2.57M | 1.00× |
| KD-SASRec | 0.7221 | 0.7782 | 4.36M | 2.32× | 0.5881 | 0.6698 | 8.64M | 2.30× | 0.4137 | 0.6174 | 0.54M | 2.79× |
| **AdaRec-SASRec** | **0.7352** | **0.7931** | **4.34M** | **6.59×** | **0.6132** | **0.6925** | **8.62M** | **5.17×** | **0.4426** | **0.6470** | **0.52M** | **3.81×** |

## 4.2 Baselines and Evaluation Metrics

To verify the effectiveness and efficiency of AdaRec, we compare it with its teacher model including NextItNet [44] and SASRec [12] which have been described in Section 3.1. In addition, we have also compared it with GRU4Rec [7] and Caser [34] for reference given that the two models are recognized as two most typical SR baselines, and SR-GNN [39], which is one of the most representative SR models with GNN. Following [43], we train Caser using the data augmentation method and train GRU4Rec based on the AR method. As for SR-GNN, we implement it as the original paper claimed.

To evaluate the recommendation accuracy, we adopt two popular top-$N$ ranking metrics, including MRR@$N$ (Mean Reciprocal Rank) and HR@$N$ (Hit Ratio) [45, 49, 18, 46]. Here $N$ is set to 5 for comparison. To evaluate the computational efficiency of AdaRec, we also compare its model size (termed as Params) and inference speedup (termed as Speedup) with the teacher models.

## 4.3 Implementation Details

We divide the user-item interaction sequence $X^u = [x^u_{1:t}]$ for each user $u$ into $X^u_{train} = [x^u_{1:t-2}]$ for training, $x^u_{t-1}$ for validation and $x^u_t$ for testing, following [12]. For the teacher model NextItNet, we set the embedding dimension $d$ to be 256, and use dilation factors of $8 \times \{1, 2, 4, 8\}$ (32 layers or 16 residual blocks). For the teacher model SASRec, we set $d$ to be 128 given that a larger $d$ hurts their performance because of overfitting. We use 8 self-attention blocks with four heads for SASRec according to its accuracy in the validation set. When searching the architecture for the student model, we set $d$ to one quarter of its teacher's embedding dimension (i.e., $d = 64$ for NextItNet and $d = 32$ for SASRec), $\gamma = 0.5$, $\beta = 8$, inner nodes $M = 3$ and student blocks $K = 4$. For training AdaRec, we employ AdamW [22] to optimize the parameters (e.g., embedding matrix and searched operations) with learning rate $\eta = 5e-3$ and weight decay of $5e-4$, and architecture distribution $P_o$ with learning rate $\eta = 2e-5$ and weight decay of $1e-4$. All the experiments are implemented in PyTorch and trained on a single TITAN RTX GPU.

## 5 Experimental Results

### 5.1 Overall Results

Table 2 reports the performance (i.e., MRR@$N$ and HR@$N$ ($N$ is set to 5), parameter size (Params) and inference speedup (Speedup)) of AdaRec and baseline models on the three datasets. From the results, we can make the following observations. First, we observe that NextItNet and SASRec outperform GRU4Rec and Caser with substantial improvements in terms of recommendation accuracy

Table 3: Performance comparison on the three datasets for cross-scenario validation by using NextItNet as the teacher model.

| Architecture | RetailRocket | | 30Music | | ML-2K | |
|---|---|---|---|---|---|---|
| | MRR@5 | HR@5 | MRR@5 | HR@5 | MRR@5 | HR@5 |
| AdaRec-RetailRocket | **0.7345** | **0.7964** | 0.6164 | 0.6956 | 0.3244 | 0.4983 |
| AdaRec-30Music | 0.7333 | 0.7953 | **0.6343** | **0.7151** | 0.3969 | 0.5951 |
| AdaRec-ML-2K | 0.7283 | 0.7926 | 0.6248 | 0.7056 | **0.4489** | **0.6519** |

Table 4: Performance comparison on the three datasets for loss ablation studies by using NextItNet as the teacher model.

| Model | RetailRocket | | 30Music | | ML-2K | |
|---|---|---|---|---|---|---|
| | MRR@5 | HR@5 | MRR@5 | HR@5 | MRR@5 | HR@5 |
| AdaRec (All) | **0.7345** | **0.7964** | **0.6343** | **0.7151** | **0.4489** | **0.6519** |
| w/o $\mathcal{L}_{KD(emb)}$ | 0.7239 | 0.7886 | 0.5976 | 0.6899 | 0.4325 | 0.6313 |
| w/o $\mathcal{L}_{KD(pred)}$ | 0.6898 | 0.7583 | 0.5512 | 0.6218 | 0.2949 | 0.4729 |
| w/o $\mathcal{L}_{KD(hidden)}$ | 0.7142 | 0.7806 | 0.6112 | 0.6981 | 0.4351 | 0.6430 |
| w/o $\mathcal{L}_{CE}$ | 0.7115 | 0.7804 | 0.5966 | 0.6883 | 0.4391 | 0.6407 |

among the three datasets, and show competitive performance with SR-GNN, which is consistent with the previous work [44, 12]. Second, AdaRec with NextItNet and SASRec as teacher models attain better recommendation accuracy than their teachers, although we do not expect AdaRec beats its teacher model in accuracy. For example, on RetailRocket and 30Music, AdaRec with NextItNet as the teacher model obtains 2.9% and 3.2% improvements over its large teacher model in terms of MRR@5. Importantly, AdaRec requires much fewer parameters and achieves notable inference speedup relative to its teachers. In addition, compared to the standard KD method [8] with equivalent model size, AdaRec with NAS techniques performs substantially better with higher inference speedup.

### 5.2 Cross-Scene Evaluation

In this section, we investigate the scene-adaptivity of AdaRec with different recommendation scenarios. We apply the searched student architecture from one recommendation scenario to other scenarios. For example, we denote the searched student architecture for RetailRocket (i.e., E-commerce domain) with NextItNet as the teacher model as AdaRec-RetailRocket, and apply it to 30Music (i.e., music domain) and ML-2K (i.e., movie domain). For such cross-scenario validation, we randomly initialize the weights of each searched student structure and re-train it using corresponding training data and the same teacher model to ensure a fair comparison. The results are summarized in Table 3, where we omit results using SASRec as teacher models due to similar behaviors. As clearly demonstrated along the diagonal line of Table 3, we can draw that AdaRec achieves the best performance on their original recommendation scenarios in contrast to other scenarios. This is, AdaRec is scene-adaptive since the searched student network only guarantees its optimal performance on a specific recommendation scenario.

### 5.3 Architecture Visualization

To better understand the basic blocks of the searched student architectures, we visualize them on the three recommendation scenarios in Figure 2. For space reason, we still only show AdaRec with NextItNet as the teacher model. By comparing the searched structures for different recommendation scenarios, we can find that AdaRec for RetailRocket (from E-commerce domain) and 30Music (from music domain) are relatively lightweight, since fewer convolution operations (i.e., $std\_cnn\_3$ for RetailRocket and $cau\_cnn\_3$ for 30Music) are used. This is likely because the two datasets have short-range sequential dependencies. On the contrary, a more complicated student structure with diverse convolution operations(i.e., $std\_cnn\_3$ and $cau\_cnn\_3$) is learned for ML-2K so as to model the long-range dependencies. The above results well back up our claim that the proposed AdaRec is able to search adaptive student structures for different recommendation scenarios.

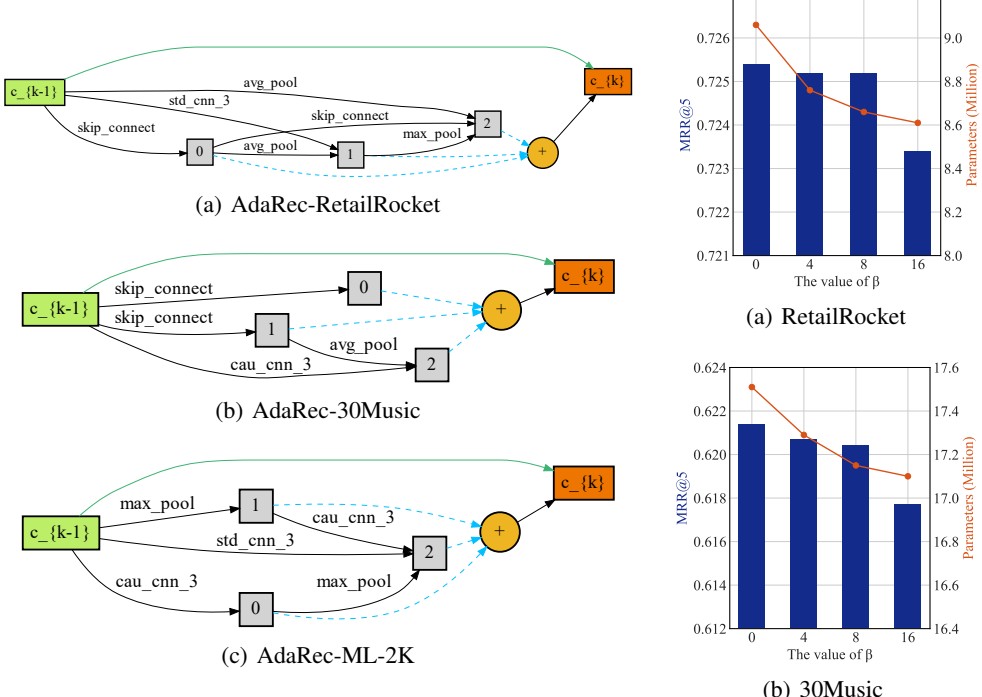

Figure 2: Visualization of basic blocks of AdaRec on the three recommendation scenarios by using NextItNet as teacher model.

Figure 3: Performance comparison on RetailRocket and 30Music for varying coefficient $\beta$ of the cost-sensitive loss ($\mathcal{L}_E$) by using NextItNet as the teacher model.

## 5.4 Ablation Studies

As described before, the loss $\mathcal{L}$ of AdaRec consists of three parts: the target-oriented KD loss $\mathcal{L}_{KD}$, the cost-sensitive loss $\mathcal{L}_E$ and the standard cross-entropy loss $\mathcal{L}_{CE}$. First, we evaluate the effects of $\mathcal{L}_{KD}$ and $\mathcal{L}_{CE}$ by removing each of them independently, as reported in Table 4. Clearly, we find that AdaRec without each of the two losses yields sub-optimal recommendation accuracy on all three datasets. Besides, it also shows that combining distillation losses on the embedding layer $\mathcal{L}_{\text{emb}}$, prediction layer $\mathcal{L}_{\text{pred}}$ and hidden layers $\mathcal{L}_{\text{hidden}}$ together produces the best results.

In addition, we verify the effect of the cost-sensitive loss $\mathcal{L}_E$ by varying $\beta$, including the default case $\beta = 8$, without constraint $\beta = 0$, weak constraint $\beta = 4$ and strong constraint $\beta = 16$. The model performance and corresponding model size are illustrated in Figure 3. From the results we can see that no constraint or a small value of $\beta$ lead to an increased model size; meanwhile, an aggressive $\beta$ results in a smaller model size but degraded model accuracy on the other hand. An appropriate constraint ($\beta = 8$) achieves the superior trade-off between the model effectiveness and efficiency.

## 6 Conclusion

In this paper, we propose a novel SR framework AdaRec based on the differentiable NAS. AdaRec compresses the knowledge of large and deep SR models into a compact student model according to their recommendation scenes. AdaRec is the first study that considers applying both KD and NAS in the SRS tasks when performing scene-adaptive knowledge compression. In addition, we devise the EMD-based KD method for effective transfer of deep hidden layers between the teacher model and the student model. A cost-sensitive constraint is introduced to achieve the trade-off between effectiveness and efficiency of SR models. Comprehensive experiments on three benchmark recommendation corpora from different scenarios show that AdaRec obtains considerably better performance compared to the standard KD baseline and its teacher model while accelerating inference time and reducing the computational workload.

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

# Appendix

## A  Preliminaries

### A.1  Task Definition

Given a sequence of user's historical behaviors $X^u = [x_1^u, x_2^u, \ldots, x_t^u]$ (interchangeably denoted by $x_{1:t}^u$), where $x_t^u$ denotes the $t$-th interacted item of user $u$, the goal of SRS is to infer the item $x_{t+1}^u$ that the user would like to interact with at time $t+1$. Since users usually pay attention to only the first few items, the top-$N$ items are recommended, referred to as the top-$N$ item recommendation problem.

### A.2  Deep Sequential Recommendation Models

**NextItNet**   The NextItNet model consists of a stack of dilated convolutional (DC) layers, which leverages a residual block to wrap every two DC layers. Specifically, each input item $x^u$ is converted into an item embedding $\mathbf{e}^u$, and the user-item interaction sequence $X^u$ is thereby denoted as the embedding matrix $\mathbf{E}^u = [\mathbf{e}_1^u \ldots \mathbf{e}_t^u]$. The embedding sequence $\mathbf{E}^u$ is then passed into a stack of dilated convolutional layers to obtain a feature vector $\mathbf{E}_l^u$ that captures the long-range historical dependencies. Here, $l$ represents the $l$-th residual block and each residual block connects two consecutive DC layers. Formally, we define the $l$-th residual block as follows:

$$\mathbf{E}_l^u = \lambda \times \mathcal{F}_l(\mathbf{E}_{l-1}^u) + \mathbf{E}_{l-1}^u \tag{18}$$

where $\mathbf{E}_{l-1}^u$ and $\mathbf{E}_l^u$ denote the input and the output of the $l$-th residual block respectively. $+$ indicates the element-wise addition. Similar to previous studies, we add a learnable coefficient $\lambda$ to the residual mappings $\mathcal{F}_l(\mathbf{E}_{l-1}^u)$, so that the model can stack more layers, and get better results than the standard version with $\lambda$ as 1. $\mathcal{F}_l(\mathbf{E}_{l-1}^u)$ denotes the residual mapping, which is defined as:

$$\mathcal{F}_l(\mathbf{E}_{l-1}^u) = \sigma\left(\mathbf{LN}_2\left(\psi_2\left(\sigma\left(\mathbf{LN}_1\left(\psi_1(\mathbf{E}_{l-1}^u)\right)\right)\right)\right)\right) \tag{19}$$

where $\psi_1$ and $\psi_2$ represent the casual convolution operations. $\mathbf{LN}_1$ and $\mathbf{LN}_2$ denote the layer normalization. $\sigma$ is the ReLU activation function.

Finally, we employ a softmax function to compute the probability distribution of the interested item $x_{t+1}^u$:

$$p(x_{t+1}^u | x_{1:t}^u) = \text{softmax}(\mathbf{W}\mathbf{E}_l^u + \mathbf{b}) \tag{20}$$

where $\mathbf{W}$ and $\mathbf{b}$ denote the learnable mapping matrix and the bias term.

**SASRec**   Similar to NextItNet, SASRec contains a stack of self-attention (SA) layers, which leverages a residual block to wrap a SA layer and a feed-forward network (FFN). Mathematically, we define the formula of the $l$-th residual block as follows:

$$\mathbf{E}_l^u = \lambda \times \mathcal{H}_l(\mathbf{E}_{l-1}^u) + \mathbf{E}_{l-1}^u \tag{21}$$

where $\mathbf{E}_{l-1}^u$ and $\mathbf{E}_l^u$ denote the input and the output of the $l$-th residual block respectively. $+$ indicates the element-wise addition. As mentioned above, We also add a learnable coefficient $\lambda$ to the residual mappings $\mathcal{H}_l(\mathbf{E}_{l-1}^u)$. $\mathcal{H}_l(\mathbf{E}_{l-1}^u)$ represents the residual mapping, which is defined as:

$$\mathcal{H}_l(\mathbf{E}_{l-1}^u) = \delta(\mathbf{SA}(\mathbf{LN_2}(\delta(\mathbf{FFN}(\mathbf{LN_1}(\mathbf{E}_{l-1}^u)))))) \tag{22}$$

where $\mathbf{FFN}$ and $\mathbf{SA}$ represent the feed-forward and self-attention operation, respectively. $\mathbf{LN_1}$ and $\mathbf{LN_2}$ represent layer normalization functions. $\delta$ is the dropout function.

Finally, we employ a softmax function to compute the probability distribution of the interested item $x_{t+1}^u$.

For both NextItNet and SASRec, we obtain the joint probability $p(X^u; \Theta)$ for the user-item interaction sequence by computing the multiplication of the following conditional distributions as follows:

$$p(X^u; \Theta) = \prod_{i=2}^{t} p\left(x_i^u | x_{1:i-1}^u; \Theta\right) p(x_1^u) \tag{23}$$

where $p\left(x_i^u | x_{1:i-1}^u; \Theta\right)$ denotes the output probability for the $i$-th item $x_i^u$ conditioned on all its previous interactions $[x_1^u, \ldots, x_{i-1}^u]$, and $\Theta$ is the set of parameters.

