# OpenReview forum: "Scene-adaptive Knowledge Distillation for Sequential Recommendation via Differentiable Architecture Search"
_NeurIPS.cc/2023/Workshop/WANT — WANT@NeurIPS 2023 Poster_

### Official Review · Reviewer_NAFZ · 2023-10-25
**The promising adaptive method of knowledge distillation for sequential recommender systems**

**Confidence:** 3

**Review:**

1. Summary and contributions:
The authors propose AdaRec, a knowledge distillation framework that uses differentiable neural architecture search (NAS) to adaptively compress the knowledge of a teacher model into a student model. The framework includes a target-oriented knowledge distillation loss to guide the search for the student network architecture, and a cost-sensitive loss to constrain model size.
The main contribution of the work is the design of the adaptive approach to choosing the architecture of the student network. In contrast to existing KD methods that distill the teacher model into a fixed-structure student model, the proposed approach doesn't fix a particular architecture of the student model.
2. Strengths:
The authors are the first to consider combining knowledge distillation and NAS in the tasks of sequential recommender systems to adaptively compress deep SR models according to their recommendation scenarios.
The proposed method is model agnostic and potentially applicable to any SR model with a deep network architecture. The authors conducted extensive experiments on three real recommendation datasets with different scenarios.
The proposed approach is compared to strong non-sequential baselines. AdaRec speeds up inference compared to its original teacher model and also achieves significantly better accuracy.
3. Weaknesses:
The paper may be difficult to reproduce as the authors don't provide a source code.
The choice of hyper-parameters for the teacher models is not clear. The authors have significantly increased the embedding dimension and the number of self-attention blocks compared to the original SASRec paper. Would be great if the authors would consider investigating AdaRec performance on teacher models with smaller parameter size.
4. Correctness:
The methodology and experimental design seem correct, but more experiments on teacher models with different parameter size would be useful to prove that AdaRec is effective for different model sizes.
AdaRec does not only speed up the inference, but also improves the quality. The authors fail to discuss this effect. It would be useful for future research to clarify the possible reasons for the quality improvement of the student network compared to the teacher network. Is this effect new or has it been found in previous work?
5. Clarity:
The paper is well written, the concept is clear. The authors provide the necessary equations and explain the methodology. The only part that was not clear was the choice of hyper-parameters of the teacher models.
6. Relation to prior work:
Authors only briefly mention existing KD methods. There is no section on related work, possibly due to space limitations.
7. Reproducibility:
The paper may be difficult to reproduce as the authors don't provide a source code.
8. Additional feedback:
The indices of the block names on fig. 2 should be fixed (c_{k-1}, c_{k})
9.  Overall score: 7

---

### Official Review · Reviewer_S3Mt · 2023-10-25
**Overall the paper is well written and offers interesting insights on applying KD and NAS to sequential recommender models.**

**Confidence:** 4

**Review:**

Summary:

The paper introduces AdaRec, a knowledge distillation (KD) framework that employs differentiable neural architecture search (NAS) to compress DNN based sequential recommender models. Although the concepts of integrating NAS and KD are well established in existing literature, this paper's primary contribution lies in their application to the field of sequential recommender models. The findings of this study demonstrate improvement in both accuracy and throughput across 3 real world datasets.

Strengths:
1. The paper is very well written and all sections are easy to follow.
2. Ablation experiments on the impact of different components of the proposed loss function and cross-scene evaluation across different scenarios provide some good insights.
3. The paper proposes an interesting approach for hidden layer knowledge distillation using the EMD algorithm to allow each layer of the student network  to learn from multiple teacher layers adaptively.

Weaknesses:
1. Related work section is missing in the paper. There are some KD and NAS approaches that have been applied to recommender systems in general (Eg: [1], [2]).
2. Since FLOPs and parameter count were used in the combined performance objective, the authors should also include FLOPs as part of the results in Table-2. Its also not clear what hardware platform was used to quantify the inference speed-up.
3. Quantification of the additional training time over-head in using the proposed approach to obtain the trained student model is missing in the paper.
4. Additionally the authors should consider including pareto frontier trade-off plots for accuracy vs parameter-count/throughput, since the constraints can be varied across different performance budgets.

References:
1. Kang, SeongKu, et al. "DE-RRD: A knowledge distillation framework for recommender system." Proceedings of the 29th ACM International Conference on Information & Knowledge Management. 2020.
2. Zhang, Tunhou, et al. "NASRec: weight sharing neural architecture search for recommender systems." Proceedings of the ACM Web Conference 2023. 2023.

---

### Official Review · Reviewer_cinu · 2023-10-25
**Decent paper that needs more clarification of evaluation and training procedure.**

**Confidence:** 5

**Review:**

The paper investigates how differentiable architecture search can be applied to deep sequential recommender networks. The proposed method allows to effectively shrink the network and even improve the recommendation quality. The paper is closely related to [Hanxiao Liu, Karen Simonyan, and Yiming Yang, 2019] with a few improvements made - the authors proposed to use additional EMD loss that helps the student model to capture more information about the teacher model, it gives the most boost in overall model performance. The authors are the first to apply this technique to the field of recommender systems. The approach described in paper allows to build models that are 5 times less in parameters size and can significantly speed up the inference time.

The quality of paper is good, the description of problem and losses is great and concise. The experiments are well-performed, the datasets have different structure and patterns.

Questions and remarks:

1. Section 2 lacks information about actual training - the proposed optimization problem is two-level, there are architecture and weights corresponding to this particular architecture. It is not clear, how the gradient step is done.
2. Eq. (1) : what is the shape of the matrix $W_2$? As I understand, the result of $W_2V+b_2$ is a matrix of shape $(n_{i}, d)$ ($n_i$ is number of items in catalogue), how do we apply softmax to it?
3. Line 138: why 1D convolutions are included into the set of operations?
4. Line 142: 'The "zero" operation facilitates the model to forget the past knowledge' - the description is vague and confusing, I suggest using 'zero operation is the absence of connection between nodes'.
5. Table 2: 'improvements of AdaRec over all baselines are statistically significant in terms of paired t-test with p-value < 0.01' - how the t-test was conducted? From the Implementation Details section 3.3 there is one validation set and one test set for all models.
6. Table 2: How many parameters do baseline models (GRU4Rec, Caser, SR-GNN) have? What is their inference time?
7. Figure 2: More description is needed. What does the green line represent, is the input of the node $c_{k-1}$ concatenated to the output of the node $c_{k}$? Are there any other intermediate operations, because sometimes the outputs of the nodes are of different size, for example Figure 2(b): the output of node 0 is bigger than output of node 2, since node 2 had pooling operation.
8. Section 4: Have you tried training the student model without the teacher model? It is interesting to show how the regular CE loss will perform (without KD loss at all).
9. What is the inference time for all datasets? The choice of ML-2K dataset is concerning, there are only 2000 users there and I think the inference time is not that big.
10. How the evaluation is performed? Are you using negative sampling? If yes, there have been good papers investigating the drawbacks of sampling during evaluation, [Krichene, W., & Rendle, S. (2020), doi: 10.1145/3394486.3403226] and [Cañamares, R., & Castells, P. (2020), doi: 10.1145/3383313.3412259].

---

### Meta-Review · Area_Chair_dcSX · 2023-10-26

**Recommendation:** Accept (Poster)
**Confidence:** 3

**Metareview:**

The reviewers are overly positive about the paper and its merits. I do agree with their assessment and recommend for acceptance.

There are however quite a few valid questions raised and I encourage the authors to address them in the updated manuscript.

---

### Decision · Program_Chairs · 2023-10-28

**Decision:**

Accept (Poster)

**Comment:**

We thank the authors for their time and contribution to WANT and we are pleased to share that after the reviewing process the paper has been accepted. Congratulations! We encourage the authors to consider reviewers' feedback for the improvement of the camera-ready version. We hope to see you in person at the workshop and brainstorm on efficient training research together!